# Olfactory Gene Families in *Scopula subpunctaria* and Candidates for Type-II Sex Pheromone Detection

**DOI:** 10.3390/ijms232415775

**Published:** 2022-12-12

**Authors:** Ting-Ting Yuan, Zi-Jun Luo, Zong-Xiu Luo, Xiao-Ming Cai, Lei Bian, Chun-Li Xiu, Nan-Xia Fu, Zong-Mao Chen, Long-Wa Zhang, Zhao-Qun Li

**Affiliations:** 1Key Laboratory of Biology, Genetics and Breeding of Special Economic Animals and Plants, Ministry of Agriculture and Rural Affairs, Tea Research Institute Chinese Academy of Agricultural Science, Hangzhou 310008, China; 2Anhui Provincial Key Laboratory of Microbial Control, Anhui Agricultural University, Hefei 230036, China

**Keywords:** transcriptomic analysis, olfactory gene, sex pheromone perception, *Scopula subpunctaria*, type-II sex pheromone

## Abstract

*Scopula subpunctaria*, an abundant pest in tea gardens, produce type-II sex pheromone components, which are critical for its communicative and reproductive abilities; however, genes encoding the proteins involved in the detection of type-II sex pheromone components have rarely been documented in moths. In the present study, we sequenced the transcriptomes of the male and female *S. subpunctaria* antennae. A total of 150 candidate olfaction genes, comprising 58 odorant receptors (SsubORs), 26 ionotropic receptors (SsubIRs), 24 chemosensory proteins (SsubCSPs), 40 odorant-binding proteins (SsubOBPs), and 2 sensory neuron membrane proteins (SsubSNMPs) were identified in *S. subpunctaria*. Phylogenetic analysis, qPCR, and mRNA abundance analysis results suggested that SsubOR46 may be the Orco (non-traditional odorant receptor, a subfamily of ORs) of *S*. *subpunctaria*. SsubOR9, SsubOR53, and SsubOR55 belonged to the pheromone receptor (PR) clades which have a higher expression in male antennae. Interestingly, SsubOR44 was uniquely expressed in the antennae, with a higher expression in males than in females. SsubOBP25, SsubOBP27, and SsubOBP28 were clustered into the moth pheromone-binding protein (PBP) sub-family, and they were uniquely expressed in the antennae, with a higher expression in males than in females. SsubOBP19, a member of the GOBP2 group, was the most abundant OBP in the antennae. These findings indicate that these olfactory genes, comprising five candidate PRs, three candidate PBPs, and one candidate GOBP2, may be involved in type II sex pheromone detection. As well as these genes, most of the remaining SsubORs, and all of the SsubIRs, showed a considerably higher expression in the female antennae than in the male antennae. Many of these, including SsubOR40, SsubOR42, SsubOR43, and SsubIR26, were more abundant in female antennae. These olfactory and ionotropic receptors may be related to the detection of host plant volatiles. The results of this present study provide a basis for exploring the olfaction mechanisms in *S. subpunctaria*, with a focus on the genes involved in type II sex pheromones. The evolutionary analyses in our study provide new insights into the differentiation and evolution of lepidopteran PRs.

## 1. Introduction

Chemical cues can function as a type of language among insects and they play a major role in insect survival and reproduction. To survive in complex and changing environments, insects have evolved a sophisticated olfactory system that allows them to effectively detect and discriminate biologically relevant pheromone compounds from a myriad of irrelevant chemicals. Major olfactory proteins include odorant-binding proteins (OBPs), chemosensory proteins (CSPs), olfactory receptors (ORs), ionotropic receptors (IRs), and sensory neuron membrane proteins (SNMPs) [1,2]. In the process of insect olfaction, odorants are bound and solubilized by soluble proteins involved in olfactory communication (such as OBPs and CSPs) and transported through the sensillar lymph, and they activate olfactory receptors (ORs and IRs) [2]. During this step, the chemical signals of the odorants are transduced into electric signals. The electric signals are transmitted to the antennal lobe, which is the primary central nervous system, through olfactory sensory neurons [3,4].

In moths, sex pheromones provide specific chemical cues and signals for mating. Based on their chemical structure, moth sex pheromone components can be divided into four types [5,6]. The components comprising straight-chain unsaturated alcohols, aldehydes, and acetate esters are classified as type-I pheromones. Type-II pheromones are straight-chain hydrocarbons with 1–3 *cis* double bonds and epoxide derivatives. Type-III pheromones are typically monomethyl- or dimethylbranched hydrocarbons with methyl branches. Type 0 pheromones are short-chain secondary alcohols or ketones. Studies on the molecular mechanisms of sex pheromone detection in lepidopteran insects have focused primarily on type I pheromones. OBPs and ORs that detect sex pheromones are named pheromone-binding proteins (PBPs) and pheromone receptors (PRs), respectively [7,8,9]. There is increasing evidence that PBPs transport pheromone molecules through the sensillum lymph to the membranes of olfactory sensory neurons, finally activating membrane-bound PRs. As well as PBPs and PRs, SNMP1 might be involved in sex pheromone detection [10].

In addition to the abovementioned genes, other soluble proteins and olfactory receptors are involved in sex pheromone detection. General odorant-binding proteins (GOBP2), a subfamily of lepidopteran-specific OBPs in *Spodoptera exigua*, *S*. *litura*, and *Plutella xylostella*, strongly bind to sex pheromones and plant volatiles [11,12,13]. Insect CSPs have various functions, including chemosensation and development. *Sesamia inferens* CSP19 and *Helicoverpa armigera* HarmCSP6 exhibit high binding affinities with their respective sex pheromone components [14,15]. IRs are expressed in antennal OSNs and act as olfactory receptors. In *Drosophila melanogaster*, IR52a, IR52c, and IR52d, comprise the so-called IR20a clade, which is required for normal sexual behavior in males [16].

*Scopula subpunctaria* (Lepidoptera: Geometridae) is a major pest affecting tea plantations in China. Its sex pheromone has two type-II components: major sex pheromone component (Z,Z)-3,9-cis-6,7-epoxy-nonadecadiene and minor sex pheromone component (Z,Z,Z)-3,6,9-nonadecatetraene [17]. Olfaction plays a crucial role in the detection ability of *S. subpunctaria*; therefore, studies on olfactory molecular mechanisms may provide potential targets for pest control. The molecular mechanisms regulating *S. subpunctaria* olfaction are currently unknown because sequence data for the olfaction genes are scarce. Moreover, the olfactory perception mechanisms of type-II pheromones are not well undestood; therefore, in this study, to identify olfactory genes, we sequenced the antennae transcriptomes of female and male *S. subpunctaria* and analyzed their phylogenetic tree, antennal abundance, and tissue expression patterns.

## 2. Results

### 2.1. Overview of Antennae Transcriptomes and Identification of Olfactory Gene Families in S. subpunctaria

The antennae of female and male *S. subpunctaria* were sequenced. After filtering and assembling, 101,626 transcripts were obtained, with average lengths of 609 bp and an N_50_ of 1016 bp. Of these 101,626 transcripts, 26,877 (26.44%) were successfully annotated using the Nr database (Appendix A). Combining the Nr annotation results and NCBI web blastx (Appendix A), 58 SsubORs were identified. Of these, 47 SsubORs had a full-length ORF. A total of 26 SsubIRs were identified, of which, 24 had full-length ORFs. Using an intact ORF, two SsubSNMPs were obtained. Forty transcripts encoding OBPs were obtained, and 11 were newly compared with those in the antennae and pheromone glands. Of the 40 SsubOBPs, 32 had full-length ORFs. From the assembled transcripts, 24 encoded CSPs and 5 SsubCSPs were newly identified. Of these, 23 contained full-length ORFs.

### 2.2. Phylogenetic Analyses of S. subpunctaria OR, IR, and OBP Genes

Phylogenetic analyses were performed by comparing the amino acid sequences of olfactory genes of *S. subpunctaria* with those of other lepidopteran insects. The OR phylogenetic tree showed that SsubOR46 was well clustered within the Orco of *Ectropis grisescens*, *Heliothis virescens*, *Bombyx mori*, and *Operophtera brumata*, with high bootstrap values (Figure 1). SsubOR13, SsubOR15, SsubOR49, SsubOR51, SsubOR52, SsubOR53, and SsubOR55 were distributed in the PR clade with the PRs of both type-I and type-II lepidopterans. SsubOR9 was grouped with EgriOR24, EgriOR31, EgriOR37, and EgriOR44, without any orthologs of other type-I lepidopteran insects. The OBP phylogenetic tree showed that SsubOBP25, SsubOBP27, SsubOBP28, and SsubOBP29 were grouped in the PBP clade with the PBPs of other lepidopteran insects (Figure 2A). SsubOBP6 and 19 were distributed in clades GOBP1 and GOBP2, respectively. Furthermore, SsubOBP25 and SsubOBP28 belonged to the PBP-A group, and SsubOBP29 and SsubOBP27 belonged to the PBP-C and PBP-D groups, respectively (Figure 2B). SsubOBP2, SsubOBP4, SsubOBP23, and SsubOBP31 belonged to the minus-C OBPs group. SsubOBP5, SsubOBP13, and SsubOBP18 were plus-C OBPs. Other SsubOBPs were determined as classic OBPs. The IR tree revealed that 10 SsubIRs (SsubIR1, SsubIR3, SsubIR7, SsubIR8, SsubIR11, SsubIR12, SsubIR16, SsubIR18, SsubIR22, and SsubIR25) were distributed in the IF75 clade (Figure 3). SsubIR4 and SsubIR24 were grouped in the IR8a/IR25a clade with other insects IR8a/IR25a. SsubIR2 and 26 were distributed in IR21a and IR40a groups, respectively.

### 2.3. Tissue Expression Profile of S. subpunctaria Olfactory Genes

The tissue expression patterns of candidate *S. subpunctaria*’s olfactory genes in the antennae were further characterized using qPCR. All SsubOR genes were expressed predominantly in the antennae and were weakly expressed in other tissues (Figure 4). Most were more strongly expressed in the female antennae than in the male antennae. Only four antenna-specific SsubORs (SsubOR9, SsubOR44, SsubOR53, and SsubOR55) were more highly expressed in the male antennae than in the female antennae. Similar to SsubORs, all SsubIRs were uniquely or highly expressed in the antennae, with higher expression levels in females than in males (Figure 5A). The qPCR results of the newly identified SsubCSPs and SsubOBPs showed that five SsubCSPs (EgriCSP20-24) were highly expressed in the legs (Figure 5B). The expression of SsubCSP19 was higher in the body than in other tissues. Ten newly identified SsubOBPs were abundant in the antennae (Figure 5C). Of these, SsubOBP32, SsubOBP33, and SsubOBP35 were expressed at higher levels in the antennae than in other tissues. Both SsubSNMPs were more strongly expressed in female than male antennae (Figure 5D).

### 2.4. Abundance of S. subpunctaria Olfactory Genes

The antennal abundance of *S. subpunctaria*’s olfactory genes was characterized by evaluating their fragments per kilobase per million mapped read (FPKM) values (Figure 6). SsubOR46 was the most abundant OR gene in the antennae of both females and males (Figure 6A). Of the 58 SsubORs, SsubOR9, SsubOR44, SsubOR53, and SsubOR55 were the four most abundant in the antennae of males, except for SsubOR46. They also showed higher transcript levels in males than in females. In the female antennae, SsubOR40, SsubOR42, and SsubOR43 were the three most abundant ORs in *S*. *subpunctaria*. Analyses of the abundance of SsubOBP mRNAs showed that SsubOBP6, SsubOBP8, SsubOBP11, SsubOBP19, and SsubOBP27 were the five most abundant OBPs in *S*. *subpunctaria* antennae (Figure 6B). Of these, SsubOBP27 was predominantly expressed in the male antennae. According to the FKPM values of SsubCSPs, SsubCSP3, SsubCSP5, SsubCSP7, SsubCSP11, SsubCSP12, SsubCSP13, and SsubCSP16 had the highest antennae mRNA abundance (Figure 6C). SsubIR4, SsubIR17, and SsubIR24 were the three most abundant IR genes in the antennae of both females and males (Figure 6D). SsubIR16 and SsubIR26 were highly expressed in the male and female antennae, respectively.

## 3. Discussion

Olfaction is critical for the survival and reproduction of insects [18]. Insects use the olfactory system to perceive host plant volatiles and sex pheromones, which subsequently drive host searching, mating, and ovipositions. To systematically study the olfactory genes in *S*. *subpunctaria*, we sequenced and analyzed its antenna transcriptome. In total, 150 candidate olfactory genes were identified; namely, 40 SsubOBPs, 26 SsubIRs, 58 SsubORs, 24 SsubCSPs, and 2 SsubSNMPs. Phylogenetic, qPCR, and mRNA abundance analyses of these genes were conducted.

The reception of sex pheromones relies largely on PRs. In lepidopteran OR phylogenetic trees, PRs for type-I pheromones are mostly grouped in the PR clade [19]. Among the 58 SsubORs, seven ORs were clustered in the PR clade, with PRs of both type-I and type-II lepidopterans; however, SsubOR9 was grouped with EgriOR31 of *Ectropis grisescens*, which was activated by two type-II pheromones, (Z,Z,Z)-3,6,9-octadecatriene and (Z,Z)-3,9-6,7-epoxyoctadecadiene [20]. Type-II pheromones are mainly produced by highly evolved insect species [6]. The OR phylogenetic tree results support this opinion; therefore, eight candidate PRs were identified based on the phylogenetic analysis results. However, according to previous studies on type-I pheromones, not all ORs belonging to the lepidopteran PR clade served as PRs. For instance, HarmOR11 and HassOR11 were identified as PR genes in *H. armigera* and *H*. *assulta*, respectively; however, HarmOR11 and HassOR11 have been silent in all compounds tested using *Xenopus* oocyte system and *Drosophila* expression system [21].

Most PRs are more highly expressed in the male antennae than in the female antennae [22]. Of the eight putative PRs identified from the OR phylogenetic tree, only four antenna specific SsubORs (SsubOR9, SsubOR51, SsubOR53, and SsubOR55) were consistent with the abovementioned characteristics; however, this number was lower than that reported for type-I and type-II moths, such as *H. armigera*, *Chilo suppressalis*, and *E*. *grisescens* [20,21,23]. Bastin-Heline et al. (2019) found that SlitOR5 from *Spodoptera littoralis* does not belong to the PR subfamily and is activated by the major sex pheromone component (Z,E)-9,11-tetradecadienyl acetate [24]. The same applies to EsemOR3 and EsemOR4; PRs of the type-0 moth *Eriocrania semipurpurella* do not belong to the moth PR clade [25,26]. These results indicate that PRs beyond the PR clade are also tuned to pheromones. In the present study, SsubOR44 was uniquely expressed in antennae, with higher expression levels in males than in females. SsubOR44 also was one of the three most abundant ORs in male antennae. Thus, we cannot rule out the possibility that SsubOR44 plays a role in sex pheromone reception in *S*. *subpunctaria*. PRs did not fall into a common clade in the moth OR phylogenetic tree, thus indicating that PRs might evolve independently within the OR repertoires of the lepidopteran order.

When co-expressed with ORs, Orco enhances odorant responsiveness without altering ligand specificity [2]. Similar to other odorants, sex pheromones are detected via heteromeric PR/Orco complexes, which operate as ligand-gated ion channels. SsubOR46 belongs to the Orco clade and is the most abundant OR gene in both female and male antennae. These results were consistent with the features of insect Orco, which are highly conserved among insect species, present in all OR-expressing OSNs, and they form heteromeric complexes with other ORs; thus, SsubOR46 might be the Orco of *S*. *subpunctaria*. As well as PRs and Orco, SNMPs, notably the SNMP1 subtype, are also involved in insect pheromone detection. In a *Bombyx mori* knockdown model, BmorSNMP1 significantly prolonged the time for males to locate mating partners [10]. In the present study, two subSNMPs were identified. Both were highly expressed in the antennae without male antenna-bias.

The external environment sex pheromone molecules enter the olfactory sensilla through cuticular pores, but they cannot directly access PRs in the dendric membrane of OSNs. Soluble proteins play roles in transporting pheromone molecules, whereas the sensillum lymph is critical to PRs [8]. PBPs are important soluble olfactory proteins that function in binding and transporting sex pheromones in both type-I and type-II moths. GOBP2 and CSP also showed a high binding affinity to type-I pheromones. Combined with our previous study findings, SsubOBP25, SsubOBP27, SsubOBP28 and SsubOBP29 are closely related to members of the lepidopteran PBP clade and SsubOBP25. SsubOBP27 and SsubOBP29 are more highly expressed in male antennae than in female antennae. As a member of the GOBP2 subfamily, SsubOBP19 is one of the most antenna abundant OBPs. SsubCSP16 was grouped into the same clade as HarmCSP6, which displayed a high binding affinity with *H. armigera* sex pheromones. These soluble olfactory proteins may function as transporters of sex pheromone components in the antennae of *S*. *subpunctaria*.

In addition to sex pheromones, ORs play an important role in detecting plant volatiles in the external environment. For example, *Manduca sexta* females preferentially oviposit on *Datura wrightii* infested with *Lema daturaphila*. *Manduca sexta* OR35 is involved in α-copaene detection [27]. In *Plutella xylostella*, OR35 and OR49 are necessary and sufficient for detecting isothiocyanates, a sulfur-containing secondary metabolite that can stimulate oviposition in female P. xylostella [28]. ORs play an important role in oviposition-site selection in female moths. In the present study, as well as the several abovementioned SsubORs, the expression of the remaining SsubORs were higher in female antennae than in male antennae. Several of them, such as SsubOR40, SsubOR42, and SsubOR43, presented higher antennal abundance. Future studies should focus on the functional characterization of these ORs.

IRs in insects serve as olfactory receptors and are classified into two subfamilies: antennal and divergent IRs [29]. Antennal IRs were first studied in *D. melanogaster* and were found to mediate the detection of certain odorants, including organic acids, amines, and aldehydes [30,31,32]. In the present study, all SsubIRs were highly or distinctly expressed in the antennae. This indicates that these IRs may be associated with olfactory functions. Moreover, SsubIR4 and SsubIR24 were clustered in the IR8a and IR25a clades, respectively, with higher antennal abundance. IR8a and IR25a function as co-receptors in moths [33]. These two SsubIRs may serve as co-receptors of other IRs.

## 4. Materials and Methods

### 4.1. Insect Samples and Tissue Collection

Adult *S. subpunctaria* individuals were first collected from a tea plantation at the Tea Research Institute of the Chinese Academy of Agricultural Sciences (Hangzhou, Zhejiang, China). Newly hatched larvae were reared in enclosed nylon mesh cages (50 cm × 50 cm × 50 cm) at a temperature of 25 ± 1 °C and a humidity of above 65% under a 14:10 h (L:D) photoperiod. After pupation, female and male pupae were maintained in separate cages. After emergence, 100 antennae of 2 day old female and male adults were collected separately for transcriptome sequencing, with two replicates. The antennae, pheromone glands, legs, heads, and bodies of 20 female adults, and the antennae, legs, heads, and bodies of 20 male adults were collected from 2 day old virgin moths for quantitative PCR with three biological replicates. All excised tissues were immediately transferred to liquid nitrogen and stored at −80 °C.

### 4.2. cDNA Library Construction, Illumina Sequencing, Assembly, and Annotation

Total RNA was extracted using the TRIzol reagent (Invitrogen, Carlsbad, CA, USA). RNA quality was assessed using gel electrophoresis and a NanoDrop One spectrophotometer (NanoDrop, Wilmington, DE, USA). The qualified RNA samples were used for cDNA library construction and Illumina sequencing with Novogene Co., Ltd. (Beijing, China), in accordance with the manufacturer’s instructions. The short-read assembly program Trinity (r20140413p1) was used for de novo transcriptome assembly, based on paired-end reads. The assembled transcripts were aligned against six databases: non-redundant database (Nr), Pfam, Clusters of Orthologous Groups (COG), Swiss-Prot, Kyoto Encyclopedia of Genes and Genomes (KEGG), and Gene Ontology (GO).

### 4.3. Identification and Phylogenetic Tree Analysis of Olfactory Gene Families in S. subpunctaria

Putative olfactory genes (OBPs, CSPs, ORs, IRs, and SNMPs) were retrieved from the NR annotation results. All retrieved sequences were confirmed using the web blastx of the National Center for Biotechnology Information (NCBI). The open reading frames (ORFs) of all putative olfactory genes were predicted using ORF Finder (https://www.ncbi.nlm.nih.gov/orffinder/ (accessed on 7 June 2022)). For phylogenetic analysis, putative OBP and CSP signal peptides were predicted using SignalP 5.0 (https://services.healthtech.dtu.dk/service.php?SignalP-5.0 (accessed on 23 June 2022)). Candidate OBPs, IRs, and ORs were aligned with those of other species using MAFFT (E-INS-I parameter). Maximum likelihood phylogenetic trees were produced using PhyML 3.1 with the LG substitution model [34]. Amino acid sequences used in phylogenetic tree analysis were listed in Appendix A.

### 4.4. Analysis of Differential Gene Expression

Transcript expression levels were calculated as FPKM using the following equation:(1)$$FPKM\left(A\right)=\frac{C\times 10^{6}}{\frac{N\times L}{10^{3}}}$$
where *FPKM*(*A*) represents the abundance of transcript A, *C* is the number of fragments uniquely aligned to transcript A, *N* is the total number of fragments uniquely aligned to all transcripts, and *L* is the number of bases in transcript A.

### 4.5. Quantitative Real-Time PCR Validation

Expression profiles of putative olfactory genes in the antennae, head, legs, pheromone glands, and bodies without antennae, were determined using qPCR on a Roche LightCycler 480 detector (Stratagene, La Jolla, CA, USA). qPCR was conducted using TB Green^®^ Premix Ex Taq (TaKaRa, Dalian, China) with a 20-μL reaction mixture containing 10 μL of TB Green Premix Ex Taq, 0.2 mM of each primer, 2 μL of cDNA template, and 6.8 μL of nuclease-free water. The reaction conditions were as follows: 30 s at 95 °C, 40 cycles of 95 °C for 10 s, and 60 °C for 30 s. Non-template reaction mixtures in which cDNA was replaced with H_2_O served as negative controls. Each reaction had three independent biological replicates and they were repeated three times (technical replicates). *Actin* and *glyceraldehyde-3-phosphate dehydrogenase* of *S. subpunctaria* were selected as reference genes [17]. Relative gene expression levels were calculated using the comparative 2^−ΔΔCT^ method [35]. Gene-specific primers are listed in Appendix A.

## 5. Conclusions

In the present study, 150 potentially olfaction genes, namely, 58 SsubORs, 26 SsubIRs, 24 SsubCSPs, 40 SsubOBPs, and 2 sSsubSNMPs, were identified in *S*. *subpunctaria*. Furthermore, candidates involved in sex pheromone reception in *S*. *subpunctaria* were obtained. The phylogenetic analysis, qPCR, and mRNA abundance analysis results of these genes provide a more comprehensive understanding of the differentiation and evolution of pheromone reception in lepidopterans. Finally, our findings enable research on the molecular olfaction mechanism of *S*. *subpunctaria*, particularly for sex pheromone reception.

## Figures and Tables

**Figure 1 ijms-23-15775-f001:**
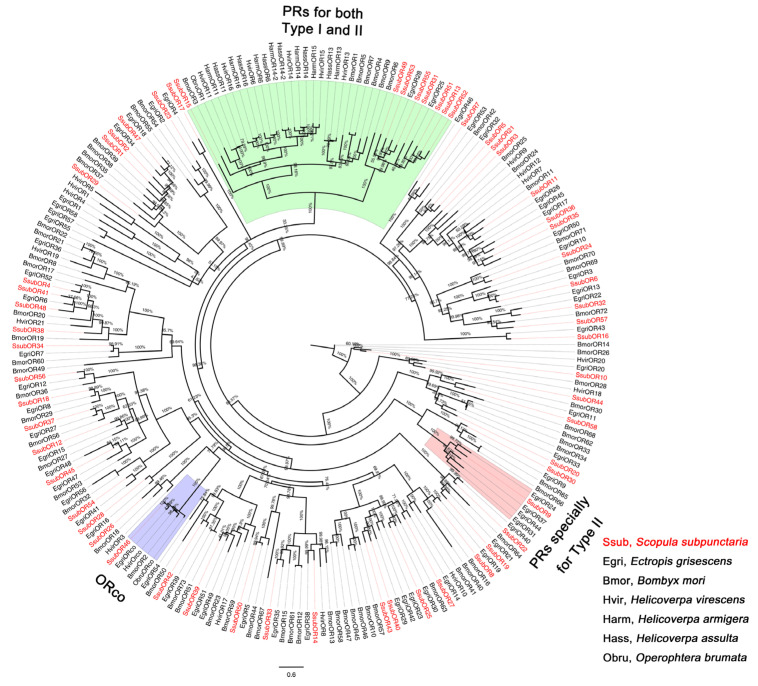
Phylogenetic analysis of SsubORs with other lepidopteran insects. The percentages on the branches show the bootstrap values. The group in purple was classified as “Orco”. The group in green was classified as “PRs for both types I and II”. The group in red was classified as “PRs especially for type-II”. The phylogenetic tree was constructed in PhyML3.1 using the maximum likelihood method.

**Figure 2 ijms-23-15775-f002:**
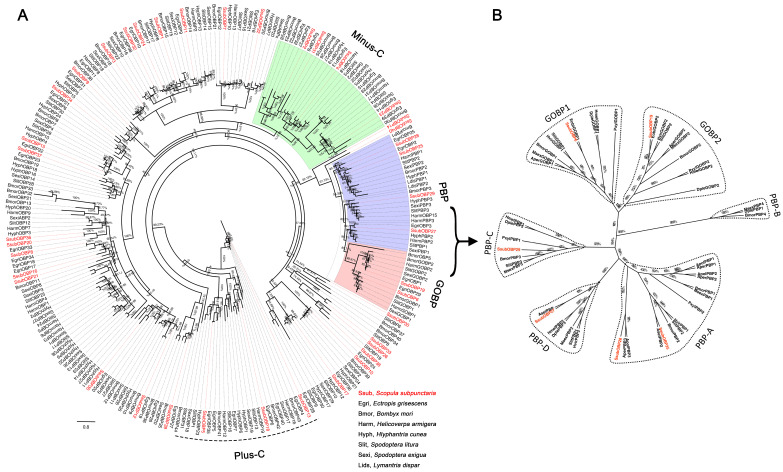
Phylogenetic analysis of SsubOBPs with other lepidopteran insects. (**A**) Phylogenetic analysis of all SsubOBPs with other lepidopteran insects. (**B**) Phylogenetic analysis of candidate SsubPBPs and SsubGOBPs with other lepidopteran insects. The percentages on the branches show the bootstrap values. The group in purple was classified as “PBP”. The group in green was classified as “Minus-C”. The group in red was classified as “GOBP”. The dashed part was classified as “Plus-C”. The phylogenetic tree was constructed in PhyML3.1 using the maximum likelihood method.

**Figure 3 ijms-23-15775-f003:**
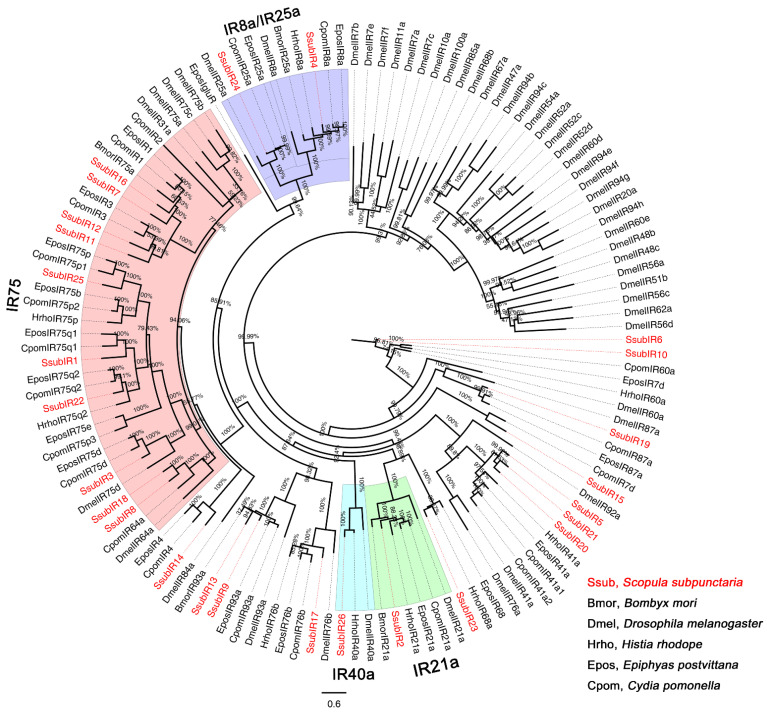
Phylogenetic analysis of SsubIRs with other lepidopteran insects. The percentages on the branches show the bootstrap values. The group in purple was classified as “IR25a/IR8a”. The group in red was classified as “IR75”. The phylogenetic tree was constructed in PhyML3.1 using the maximum likelihood method.

**Figure 4 ijms-23-15775-f004:**
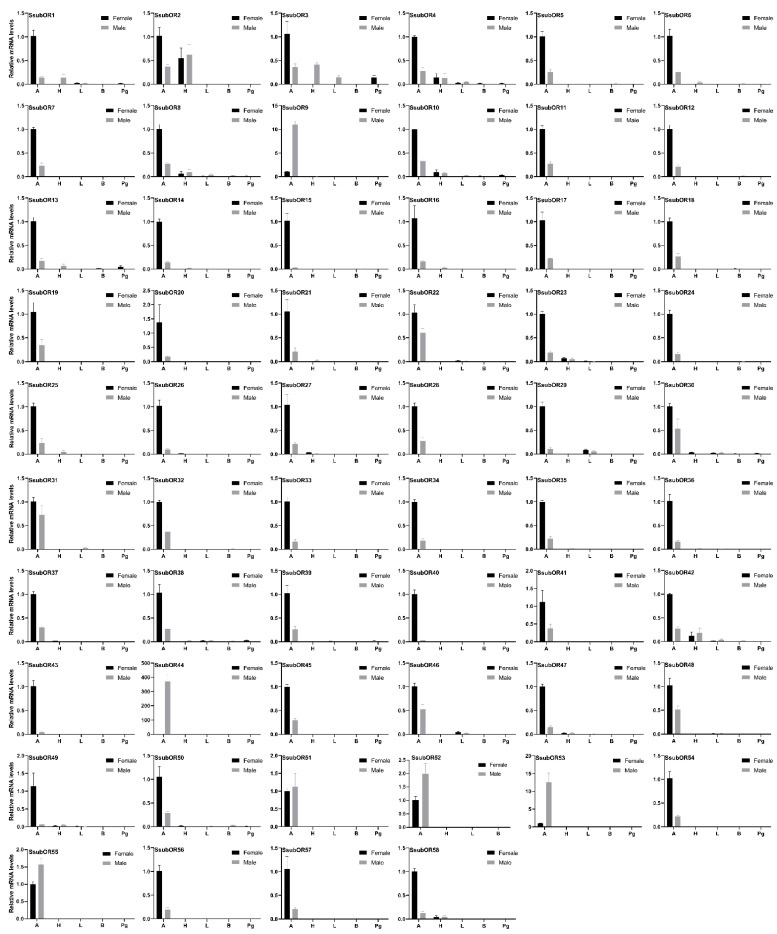
Tissue expression profile of SsubORs. A, antenna; H, head; B, body without antennae, head, legs, or pheromone gland; L, leg; Pg, pheromone gland. The black columns represent female moths. The grey columns represent male moths. The expression levels are relative to those in female antennae.

**Figure 5 ijms-23-15775-f005:**
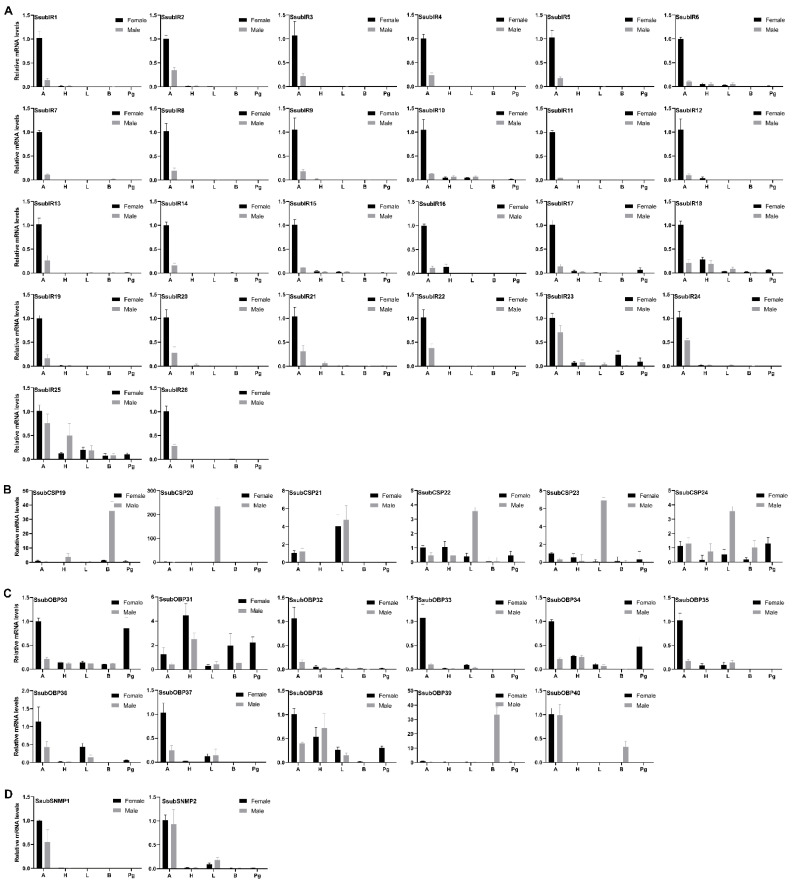
Tissue expression profile of SsubIRs, SsubCSPs, SsubOBPs and SsubSNMPs. (**A**): Expression profiles of SsubIRs. (**B**): Expression profiles of SsubCSPs. (**C**): Expression profiles of SsubOBPs. (**D**): Expression profiles of SsubSNMPs. A, antenna; H, head; B, body without antennae, head, legs, or pheromone gland; L, leg; Pg, pheromone gland. The black columns represent female moths. The grey columns represent male moths. The expression levels are relative to those in female antennae.

**Figure 6 ijms-23-15775-f006:**
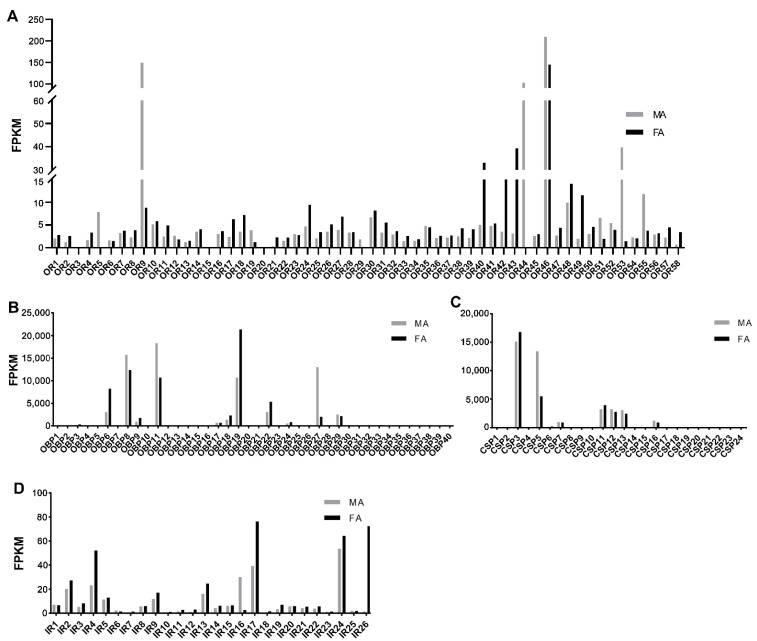
Expression levels of ORs, OBPs, CSPs, and IRs in *Scopula subpunctaria* antennae based on the FPKM values. (**A**): FPKM values of ORs in *S. subpunctaria* antennae. (**B**): FPKM values of OBPs in *S. subpunctaria* antennae. (**C**): FPKM values of CSPs in *S. subpunctaria* antennae. (**D**): FPKM values of IRs in *S. subpunctaria* antennae. FA, antennae of female individuals; MA, antennae of male individuals.

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
