# Peer review of "Olfactory Gene Families in Scopula subpunctaria and Candidates for Type-II Sex Pheromone Detection"

_ijms, 2022, doi:10.3390/ijms232415775_

Round 1
Reviewer 1 Report
The paper entitled “Olfactory gene families in Scopula subpunctaria and candidates for type-II sex pheromone detection”, highlights the profile of olfactory genes presented in S. subpunctaria, their phylogeny and expression profiles. Overall, the article is well written and the methodologies seem well performed. To me, this article could be potentially published. However, authors must address the following aspects before publication:
1. “SNMP1 captures pheromone molecules from PBP and delivers them to associated PRs [10]” (see introduction section). This part should be written with much more caution. Although BmorSNMP1 interacts with pheromone receptors in yeast hybridization studies, but there still lacks in vivo evidence.
2. The authors identified four putative PBPs from S. subpunctaria. However, the phylogenetic tree did not provide a clear relationship between PBP-A, PBP-B, PBP-C, and PBP-D groups (Vogt, et al. Insect Biochemistry and Molecular Biology 2015, https://doi.org/10.1016/j.ibmb.2015.03.003).
3. Please explain exactly what constitutes a biological replicate in the qPCR experiments. Was one cDNA sample produced from each tissue and tested on three different occasions? Were three different cDNA samples produced from each RNA extraction? etc. Also in the qPCR experiment, the relative expression levels are relative to what exactly? To the tissue with the lower expression levels? To one of the tissues?
4. Is there any evidence that IRs recognize sex pheromones in lepidopteran species? If so, please provide reference.
5. In the discussion section, the authors attempt to explain the function of the receptors based on sex and tissue-biased expression patterns. While expression differences may be real, correlating these to function isn’t appropriate based on what is known about the chemical ecology of moths as well as the specificity of receptors. The authors make it sound like only males should express pheromone-specific ORs (PRs) and only females should express ORs for plant volatiles. However, it is very common for males to detect the same plant volatiles that females use as oviposition cues, and also for females to be able to detect sex pheromone molecules.
6. Line 11-12, “produce” should be “produces”, “which are” should be “which is”
Line 190-191, “Xenopus” and “Drosophila” should be italicized
Author Response
Dear reviewer:
We thank you for your thoughtful suggestions and insights. The manuscript has benefited from these insightful suggestions.The responses to all comments have been prepared and attached herewith/given below.
I look forward to hearing from you.
Sincerely,
Zhao-qun Li

Reviewer 2 Report
The manuscript entitled "Olfactory Gene Families in Scopula subpunctaria and Candidates for Type-II Sex Pheromone Detection" has shown the transcriptome research about the olfactory of a major tea pest, Scopula subpunctaria, in China. It provides useful information on the olfactory detection of type-II sex pheromone components. The work sounds interesting, but this manucript has many logical and careless mistakes, and lack of introduction and explanation of results. The following are the questions and some mistakes in this manuscript:
1. In Lines 88-89, "average length of 659 bp, an N50 of 2500 bp", but in the supplementary files 1, "mean length of 609bp, N50 of 1016bp" can be found. Which one is the final information?
2. In the whole text, I cannot find the information discussed about Supplementary File 2. Is there any missing part?
3. In Lines 90-91, the authors should refer to Supplementary File 1, and Supplementary File 4 or 5 clearly, not just refer to "supplementary files".
4. In Result part 2.1, The identification of the olfactory genes was not elucidated enough. For example, Lines 91-92, "47 SsubORs had a full-length ORF with characteristics typical of insect OR genes", the authors should show the characteristics struture of the typical OR gene on details. It would help readers understand the logical flow.
5. All the legends of figures for phylogenetic trees were not precise and have many missing indications. For example, the authors did not show the exact bootstrap values for the phylogenetic tree, but in Line 103, the authors said "high bootstrap values". And in the group of "high bootstrap values", I find a bootstrap value of 41.09% included. But the authors did not make more explanations. Even there are many missing indications, including the group in green, the group in purple, the group in red, and so on. Some misunderstandings due to the missing information. For example, Lines 104-105, "SsubOR13, 15, 49, 51, 52, 53, and 55 were distributed in the PR clade with the PRs of both Type I and Type II lepidopterans". Is there other clade with PRs of both Type 1 and Type II or only one clade with the PRs of both Type 1 and Type II in figure 1? And the authors did not list the homologous olfactory genes of other insects as a part of supplementary files.
6. In Lines 129, SsubOR 52 also showed more highly expressed in male antennae than in female antennae. But it was not included in antennae-specific SsubORs. Why?
7. The legends of figure 4 and figure 5 are not accurate. For example, the authors indicate MA as antennae of male individuals, but in the figures, only "A" was shown.
8. In the Result part of 2.3, why the qPCR results of SsubCSP1-18 and SsubOBP1-30 were not included in figure 5 ?
Lines 135-136, as authors showed "SsubOBP32, 33, and 35 were expressed at higher levels in the antennae than in other tissues", why they did not include SsubOBP30, 36, 37, and 40 ?
9. In the Result part of 2.4, the authors showed the abundance of the olfactory genes by FPKM values. Did the authors compare it with the results from qPCR? Did they match quite well?
Author Response

(The authors gave the same response as above.)
